# Analysis of the Relationship between Tobacco Smoking and Physical Activity in Adolescence: A Gender Specific Study

**DOI:** 10.3390/medicina57030214

**Published:** 2021-02-27

**Authors:** Dora Maric, Antonino Bianco, Ivan Kvesic, Damir Sekulic, Natasa Zenic

**Affiliations:** 1PhD Program in Health Promotion and Cognitive Sciences, Sport and Exercise Research Unit, Department of Psychological, Pedagogical and Education Sciences, University of Palermo, 90144 Palermo, Italy; dora.maric@unipa.it (D.M.); antonino.bianco@community.unipa.it (A.B.); 2Faculty of Science and Education, Mostar, Bosnia and Herzegovina, University of Mostar, 88000 Mostar, Bosnia and Herzegovina; ivan.kvesic@fpmoz.sum.ba; 3Faculty of Kinesiology, University of Split, 21000 Split, Croatia; dado@kifst.hr

**Keywords:** tobacco, physical activity, puberty, predictors, risk factors

## Abstract

*Background and Objective*: Although smoking and the physical activity level (PAL) are important determinants of health status in adolescence, there is a lack of information on the relationship between smoking and PAL in early adolescence. The objective of this study was to evaluate the gender-specific relationship between smoking and PAL in 14-to-16-year-old adolescents. *Materials and Methods*: The sample included 650 adolescents (337 girls, 14.7 ± 0.5 years at first testing wave) from Bosnia and Herzegovina. During the first testing wave, participants were tested using structured questionnaires. Second testing was commenced after approximately 20 months (16.4 ± 0.6 years). The variables were age, gender, socioeconomic status, living environment, cigarette smoking (predictors), and PAL (criterion). Predictors were measured at the first wave, and criterion at the first-wave and second-wave. *Results*: For girls, smoking was negatively correlated to PAL at the first-wave (OR: 0.75, 95% CI: 0.55–0.95) and at the second-wave (OR: 0.73, 95% CI: 0.71–0.96). No significant association between smoking and PAL was found for boys. Results suggest that adolescent boys and girls do not follow the equal trajectories when it comes to relationships between smoking and PAL. *Conclusions*: In developing promotional public health actions related to a decrease of smoking and increase of PAL, a gender-specific approach is highly recommended. Further studies analyzing the cause–effect relationship between consumption of other types of psychoactive substances and PAL in this age group are warranted.

## 1. Introduction

Physical activity (PA) includes every body movement, including time spent on activities such as transportation, household chores, work, and recreational and sports activities. It is categorized according to intensity levels, from low to moderate to vigorous [1]. The PA is considered to be any movement performed by muscle contraction, which results in an increase in energy expenditure or an increase in the metabolic rate [2]. Due to its direct and indirect influence on health status globally, the World Health Organization (WHO) proclaims PA as a health imperative, and makes it one of the key priorities for optimizing health and well-being [1]. 

The health-promoting effects of PA are backed by numerous studies exploring the effect of PA on the prevention and treatment of chronic diseases [3,4,5]. Some of the known benefits of regular and adequate PA are: improvements of bone and functional health, improvements of muscular and cardiorespiratory fitness, and reduced risk of coronary heart disease, hypertension, stroke, various types of cancer, diabetes, and depression. It is fundamental for weight control and energy balance [1,6,7]. Furthermore, scientific evidence highlights the benefits of PA in a broader socio-economic context, given that people who are physically inactive due to a wide range of health consequences represent a great burden for the health care system and, thus, a large financial expense for a state [8]. 

Despite the persistent and continuous promotion of guidelines for maintaining and optimizing health status over the last two decades, the trend of physical inactivity continues to increase rapidly on the global level [9]. The decline in physical activity levels (PAL) is largely due to the modern way of life, which is strongly influenced by rapid economic development and changes in the living environment. Specifically, the increase of sedentarism and screen time, lack of active transportation, and the low necessity of physical work in everyday life are the most recognizable reasons for a decrease of everyday PAL [10]. However, the fact that a rapid decline in PAL occurs in early adolescence is particularly concerning. On a global level, 81% of adolescents (84% of girls, 78% of boys) aged 11 to 17 years failed to meet the WHO recommendations in 2016 [1]. It is important to emphasize that adolescence is a crucial period for the adoption and retention of future habits, and, thus, it largely determines the future of the individual. Therefore, it is reasonable to conclude that physical activity in adolescence determines later levels of activity, and, thereby, directly affects the overall health status of individuals in the future [11]. As a result, it is necessary to evaluate factors influencing PA in this population in order to design more effective interventions [11]. One of the factors, which is particularly important with regard to PAL in adolescence is substance misuse, including tobacco smoking [12,13,14,15]. 

The smoking prevalence is generally decreasing globally. However, smoking is still a sixth leading cause of death worldwide, and the most frequently misused psychoactive substance in the world, despite the fact that the causal link between lung cancer and smoking was established more than 50 years ago [16,17,18]. Statistics and international data show Southeastern Europe (including Bosnia and Herzegovina) as one of the territories with the highest prevalence of adolescent smoking in Europe [19]. In particular, for Bosnia and Herzegovina, daily smoking is reported for >25% of high school adolescents, while an additional 20% of adolescents misuse cigarettes but not on a daily basis [20]. This is explained by several facts including traditional orientation of tobacco growing in some parts of the country, as well as relatively low prices of cigarettes (pack rarely costs more than 3 Euros). Consequently, cigarette smoking is socially accepted in the country, and in the whole region (i.e., the territory of former Yugoslavia) [21,22,23]. 

Knowing that both smoking and PAL are important determinants of health status, studies have analyzed the smoking prevalence as correlating to PAL in adolescence [20,22,23]. Although PA and participation in sports (as the most important determinant of PAL in adolescence) are traditionally considered a way of supporting healthy habits in adolescence, and, therefore, higher PAL should be correlated with lower smoking, current research does not always support this thesis [24]. The research conducted on high school students in South Carolina suggests that higher levels of physical activity are correlated with lower likelihood of cigarette smoking in male adolescents [15], which is consistent with other studies done on adolescents that addressed this issue [12,25,26]. Conversely, an African study conducted on children aged 13 to 15 years found no significant association between cigarette smoking and PAL [27]. Gender-specific associations between smoking and PAL were also detected [28,29]. Observed inconsistencies among studies are expected since it is well established that studies investigating factors associated with PALs are indicating differences among different populations depending on the sociocultural environment and gender caused by different confounding effects on established relationships between studied variables. However, most of these studies observed associations only in one time-point, and such a design provides limited evidence on the relationships, which may exist between smoking and PAL in adolescence.

Irrespective of the associations between smoking and PAL, the causality between these variables can be interpreted in both directions. More specifically, the smoking can be observed as “the cause” of lower PAL. For example, the higher smoking negatively influences the physical capacities and, therefore, reduce adolescents’ willingness to participate in physically demanding activities. This is clearly shown in studies where authors confirmed that adolescent smokers were more likely to have quit sports than non-smokers [30]. Meanwhile, it is possible that PAL actually influence the smoking. This is based in the socio-psychological theory of self-categorization, which states that people accept and adopt the beliefs, behaviors, and norms of members of their social-environment [31]. As a result, if adolescents embrace the socially acceptable, healthy behaviors in childhood (including higher PAL due to participation in sports or leisure time PA), it will probably transfer in lower smoking [32]. Although both mentioned directions of cause-effect relationships deserve attention, for a moment, we were particularly interested in the possible influence of smoking as a predictor of PAL. 

The rapid decline in PAL and the increase in smoking that occur in adolescence indicates the need for prospective studies that would give us a better perspective and a more comprehensive interpretation of the relationship and the cause–effect relation of these variables. To the authors’ knowledge, there is only one prospective study done on the territory of Southeast Europe that investigated smoking as a predictor of PAL during adolescence [33]. The study reported no significant influence of smoking on PAL in the period between 16 and 18 years of age. However, given that most of the studied adolescents initiated smoking before the age of 16 (the age of study baseline), the authors of that study highlighted: (i) the need for further research examining younger children, and (ii) gender-stratified analyses due to significant differences in smoking prevalence and PAL between genders [33]. Therefore, the aim of our study was to examine the gender-specific relationship between smoking and PAL in adolescents aged 14 to 16 years. We hypothesized that smoking will be negatively correlated to PAL at the first testing wave (beginning of high school education, 14 years of age), and at the second testing wave (end of second year of high school, 16 years of age) both in boys and girls. 

## 2. Materials and Methods

### 2.1. Participants and Design of the Study

This study is part of a larger investigation where different types of predictors were observed in relation to PAL in adolescents from Bosnia and Herzegovina, and results of other analyses with other samples of participants are recently published elsewhere [34]. In this investigation, we observed 650 participants (337 (48.2%) females) aged 14.7 ± 0.5 years at the first testing wave. The sample included adolescents from Herzegovina Neretva, Western Herzegovina, and Herzeg Bosnian County/Canton 10 (Figure 1). For the purpose of the investigation and this study, we used a multi-stage cluster sampling method, including (i) random selection of one-quarter of high schools in the observed Cantons and (ii) random selection of one-third of the first-year classes. With the population of 2662 first-year high-schoolers, previously reported prevalence of smoking for somewhat older adolescents (30%), Type I/II error rate of 0.05, and statistical power of 80%, the necessary sample size was 311 participants. In all schools observed in this study, children participated in physical education classes twice a week during the school year, which was a mandatory school requirement for all healthy children. Some children participated in out-of-school sports, and other physically demanding activities and programs (i.e., dance), but, for the purpose of this investigation, these data were not collected (please see more for details).

The sample was tested at two testing waves. First testing wave was done when participants were 14 years old on average and started with their second-wave (approximately 20 months later). At the first wave, 699 participants were tested. However, only those participants who were tested at both testing waves were included in the study. Prior to investigation, this study was approved by the ethical board of each corresponding author’s institution, and then authorized by school authorities. The location, time frame, and number of participants across testing waves is presented in Figure 1.

Before the first-wave testing, a consent form was collected from children’s parents/responsible adults. During both testing scenarios, the participants were informed that the survey was strictly anonymous, that they can refuse to participate, and that they could leave some questions or the entire questionnaire unanswered. The participants were asked to use self-selected codes in order to track the responses. The testing was performed through an Internet-based application. Testing was done in schools, using the originally designed online survey. Participants used their private smart phones, but one of the investigators was present at each testing site and provided a smart phone if it was necessary.

### 2.2. Variables

Participants were tested using the structured questionnaires that were previously found to be reliable and valid measuring tools: the Questionnaire of Substance Use (QSU) and the Physical Activity Questionnaire for Adolescents (PAQ-A) [35,36].

The QSU evaluates sociodemographic factors, and data on substance misuse, including smoking prevalence. The sociodemographic factors included the participants’ age (in years), urban/rural living environment (later derived on the basis of specified living community), gender (male—female), and socioeconomic status of the family (SES, above average—average—below average). One query was used to evaluate prevalence of cigarette smoking with four possible answers (never smoked, ever tried, smoking but not daily, and daily smoking). For the purpose of this study and later statistical analyses, answers were grouped in nonsmokers vs. smokers [32,35]. 

PAQ-A was used for PAL assessment. The PAQ-A consists of nine items. In brief, participants are asked to report their PAL based on the last seven days (one week). The first eight items are scored on a 5-point scale and include questions on different types of PAL (i.e., active transportation, activity during physical education classes, free play, sports), and, as a result, the PAQ-A provides a summary PA score derived from eight items. The ninth item does not contribute to the total score. It is used simply for selection (i.e., illness or injury). The final score at PAQ-A ranges from 0 (minimal score) to 5 (maximal). For the purpose of this study, we observed the total score of the PAQ-A, but additionally categorized data as “insufficient PAL” (scores < 2.73), and “sufficient PAL” (scores of 2.73 and above) [36,37].

### 2.3. Statistics

To test the normality of the distributions, the Kolmogorov–Smirnov test was calculated. For numerical variables (age, PAQ-A raw scores), the means and standard deviations were calculated, while frequencies and percentages were reported for the remaining variables.

The *t*-test for independent samples was used to identify differences between groups in parametric variables (PAQ-A raw scores, age). The χ^2^ test was used to identify differences between groups in ordinal variables, and to test the associations between ordinal variables (i.e., gender and PAQ-A observed at a categorical scale (sufficient vs. insufficient PAL)). Changes in PAQ-A raw scores during the study course were evaluated by a *t*-test for dependent samples. 

Binary logistic regression was used to estimate relationships between studied predictors observed at the first-testing wave and dichotomized PAQ-A criteria (e.g., insufficient vs. sufficient PAL, please see previous text for details), and the odds ratio (OR) and the corresponding 95% confidence intervals (95% CI) were reported. Since preliminary values indicated a strong influence of gender on PAL, the logistic regressions were controlled for gender as a covariate. Finally, gender was observed as an effect modifier, and all variables were additionally checked for correlations in logistic regressions stratified for gender. Hosmer Lemeshow test (HL) was calculated to test the model fit. 

Statistical significance was set at 95%, and all analyses were done by software Statistical ver. 13.5 (Tibco Inc., Palo Alto, CA, USA).

## 3. Results

The 80.1% of boys and 80.9% of girls never tried to smoke cigarettes, with no significant difference between genders (χ^2^ = 0.26, *p* = 0.60). 

Figure 2 presents PAL at testing waves. PAQ-A decreased significantly for a total sample (2.40 ± 0.81 vs. 2.25 ± 0.77, *t*-test = 22.19, *p* < 0.001), for boys (2.56 ± 0.86 vs. 2.34 ± 0.81, *t*-test = 16.75, *p* < 0.001), and for girls (2.25 ± 0.76 vs. 2.15 ± 0.72, *t*-test = 13.04, *p* < 0.001).

Adolescents who had sufficient PAL were of a similar age as their peers with insufficient PAL at the first-wave (14.79 ± 0.22 and 14.81 ± 0.42 years, respectively, *t*-test = 1.24, *p* = 0.21). At the second-wave, those adolescents who achieved sufficient PAL were younger than those with insufficient PAL (14.76 ± 0.42 and 14.83 ± 0.41, respectively, *t*-test = 2.26, *p* = 0.03).

Differences between adolescents according to their PAL observed at a categorical scale (e.g., insufficient vs. sufficient PAL) at the first-wave are presented in Table 1. Regarding the *t*-test where genders were compared in PAQ-A raw scores (please see previous results of the *t*-test for an independent sample), boys are more likely to achieve sufficient PAL than girls at the first testing wave (χ^2^ = 7.55, *p* < 0.001). 

Differences between groups of adolescents according to their PAL observed at a categorical scale (insufficient vs. sufficient PAL) at the second-wave are presented in Table 2. As could be expected from *t*-test results (please see previous text), boys are more active than girls (χ^2^ = 7.54, *p* < 0.001), while smoking is lower in those adolescents who achieved sufficient PAL (Mann Whitney test = 6.46, *p* = 0.02).

Logistic regression calculated for “Gender” as a predictor of PAL showing a higher likelihood for sufficient/appropriate PAL among boys at the first-wave (OR = 1.68, 95% CI = 1.21–2.35), and at the second-wave (OR = 1.59, 95% CI = 1.12–2.26).

Table 3 presents correlations between studied predictors and PAL at the first-wave. When calculated for the total sample, while including “Gender” as a confounding factor, the lower likelihood for sufficient PAL was found for adolescents who smoke (OR = 0.55, 95% CI: 0.36–0.83, HL = 0.56, *p* = 0.90). In gender-stratified models, a significant relationship was found between smoking and PAL in girls, with lower PAL among girls who smoke (OR: 0.75, 95% CI: 0.55–0.95, HL = 0.04, *p* = 0.84). 

Correlates of PAL at the second-wave are presented in Table 4. Smoking was correlated to PAL (OR = 0.59, 95% CI = 0.37–0.92, HL = 0.13, *p* = 0.98), with a lower likelihood for sufficient PAL in adolescents who smoke (for the total sample with “Gender” as a confounding factor). When gender-stratified logistic regression were calculated, the lower likelihood for sufficient PAL was demonstrated in girls who smoke (OR = 0.73, 95% CI: 0.51–0.96 HL = 0.12, *p* = 0.72). 

## 4. Discussion

The study aimed to examine the impact of smoking on PAL in adolescents 14 to 16 years of age. With regard to the study’s aims, we can highlight the following results. First, male adolescents were more physically active than females. Next, cigarette smoking negatively affects PAL in girls. Finally, no association between smoking and PAL was found in boys. Therefore, our initial study hypothesis can be only partially accepted. 

### 4.1. Gender and Physical Activity Levels in Adolescence

The decline in PAL in adolescents is not surprising and is consistent with a large number of previous studies [38,39,40]. This phenomenon is explained by numerous factors, including changes in life priorities and time requirements (longer sedentary time at school and home), focusing on academic achievement [41], and a lack of support from friends, teachers, and families [42,43]. Furthermore, it is generally believed that the decline in PAL is largely due to the modern way of life, which is strongly influenced by rapid economic development and changes in the living environment. Finally, it is important to emphasize that the decrease of PAL in this period is significantly influenced by cessation of organized sports [33].

We verified lower PAL among girls than in boys at both testing waves, and this is in accordance with studies done globally [44]. We can assume that this is actually a result of a significant decrease of PAL among girls in earlier age, given that the changes caused by puberty occur earlier in girls [38]. Although the authors are currently unable to offer clear evidence for such a statement, we may support it based on previous research that states that biological factors significantly contribute to gender differences in PAL [40,45,46]. In short, differences in PAL between boys and girls decrease after adjustment to sexual maturity, in which case, we can assume that lower PAL in girls could be associated with maturation at an earlier chronological age [47,48]. 

Another explanation for higher PAL in boys is related to sport participation. Although, in this study, we did not specifically observe the level of sport participation, recent studies with somewhat older adolescents in the region showed a significant influence of participation in organized sports as an important determinant of PAL in adolescents [49]. In addition, investigations regularly confirmed higher sport-participation in adolescent boys than in girls [23,50]. Together with previously mentioned differences in maturation stages between boys and girls, this may result in gender-differences in PAL we verified in this study.

### 4.2. Smoking and Physical Activity Levels in Girls and Boys

Research to date brought inconsistent results when it comes to association between smoking and PAL in adolescents [15,24,51]. For instance, the US study with high school students showed higher PAL in male adolescents who do not smoke [15], and similar findings are reported in some other studies [12,25,26]. In the meantime, an African study reported no significant association between PAL and cigarette smoking [27], which is consistent with a recent report from Bosnia and Herzegovina with older adolescents [33]. As a result, we may say that gender-specific associations between smoking and PAL shown herein are not surprising. That being said, we may assume that differences in results occur due to dissimilarities among populations, which leads to the occurrence of different confounding effects on established relationships between studied variables. Accordingly, we discussed our results in the context of gender specifics and the socio-cultural environment of the tested sample.

In our study, cigarette smoking was negatively correlated with PAL in girls, and we can highlight several explanations for such a finding. First, the negative effects of smoking on the cardiovascular and respiratory systems are well known. Smoking leads to a decrease in oxygen uptake and transport, which leads to the reduced functionality of the cardiovascular and respiratory systems during exercise [15]. Negative effects that often manifest immediately, such as breathing difficulties, are logically more familiar to physically active individuals, which explains the association between lower smoking and higher PAL among girls in our study.

It is important to highlight that such a negative relationship between smoking and the ability to be physically active is particularly evident for girls. Previous experimental studies confirmed a dose-response relationship between smoking and reduced lung function (i.e., lower physical capacity) only in girls [52]. It actually means that girls should be observed as more vulnerable than boys in terms of the impact of smoking on lung function and respiratory symptoms, which additionally may explain the established relationship between smoking and PAL (exclusively) in girls. It altogether likely resulted even in a relatively consistent relationship between smoking and PAL in girls.

The smoking status observed as the first testing-wave was not significantly related to PAL in boys. Although this result is in certain disagreement with findings on higher PAL among girls who do not smoke, it can be explained when taking into account the context of PAL in studied boys. Previous studies noted that PAL in Bosnian and Herzegovinian boys is mostly a result of their participation in competitive sports (i.e., soccer, handball, and basketball) [32,36]. Meanwhile, this is not the case for adolescent girls from the same country since girls’ PAL is more a result of participation in recreational activities in fitness centers, and in nature (jogging, walking, etc.). It is also important to note that girls’ participation in competitive sports is much lower than in boys of the same age [53]. 

The explanation for such gender-specific context of the PAL during adolescence is related to the existence of generational, cultural, and social influences on sports’ involvement in men and women [41,54]. Briefly, a competitive sport is presented as an important determinant of masculinity in many cultures, and, thus, the social expectation for men to play sports is higher than that for women [41]. Accordingly, boys unite their sports’ identity with their masculine identity relatively easily, which is not easy for the girls who often lack the support of their social environment (parents, teachers, and friends) [55]. Finally, studies clearly indicate that the competitive nature of the sport is unattractive to a great number of females who often feel that they do not have the adequate level of competence required to participate in a team [42,56]. 

Furthermore, previous investigations in the region noted alarmingly high prevalence of smoking in boys who participate in organized competitive sports (among athletic boys aged 16–18 years more than 30% were daily smokers), and studies indicated even the earlier initiation in smoking in athletic than in non-athletic boys [20,23]. This was explained as the socio-cultural phenomenon of competitive sport, especially regarding the regular post-sport gatherings in bars and clubs where smoking is allowed. It actually could result even in “negative” correlations between smoking and PAL even in observed boys (i.e., boys who smoke are more likely to be involved in sports, and, consequently, will have higher PAL). 

Collectively, all previously discussed issues likely led to a non-significant relationship between smoking and PAL among boys in our study. Figuratively speaking, while the higher PAL should be logically influenced by non-smoking (and will result in a correlation similar to one experienced among girls), the specific context of previously explained factors (post-sport gatherings, high-prevalence of smoking, and early initiation in smoking among athletic boys) caused the opposite relationship, finally resulting in a non-significant relationship between smoking status, and PAL among adolescent boys observed herein.

Finally, there is an additional possibility that the lack of association between smoking and PAL in boys could be aggravated by low vulnerability of boys when it comes to a negative impact of smoking on their physical capacities [52]. Specifically, in the study which examined a sex-specific effect of adolescent smoking on respiratory symptoms and lung function, authors confirmed the dose-response relationship between smoking and reduced lung function only in girls [52]. In other words, boys were generally more resistant to the impact of smoking on respiratory symptoms and lung function. Together with all previously specified factors, this could partially influence our findings as well. 

### 4.3. Limitations and Strengths 

The most important limitation of our study is the fact that this study is based on self-reports. This means that PALs and cigarette smoking were not directly and objectively measured. However, they were demonstrated throughout the participants’ self-report. The questionnaires were used because of a relatively large sample of participants, testing of the cigarette smoking data, and a repeated measurement study design, which, altogether, limited the usage of more accurate measurement tools (i.e., accelerometers and pedometers). However, used questionnaires were reported to be valid and reliable measuring tools. Moreover, even though there is a certain probability that participants may choose socially desirable answers, particularly when it comes to cigarette smoking, authors believe that strict anonymity of the investigation and previous experience from similar studies conducted in the past are decreasing this possibility. Next, in this investigation, we observed only tobacco smoking, and no information was provided about other types of smoking (i.e., e-cigarettes, water pipe). However, knowing the situation in the region where the study was performed, and relatively low prevalence of types of smoking other than “cigarette smoking,” we believe that this limitation did not influence our findings to a greater extent. Furthermore, our study lacks the detailed evaluation of smoking history and is no cohort study following individual changes of smoking and PAL, but only a repeated cross-sectional study in an aging adolescent population. Prospective cohort approaches should be considered in future research.

This is one of the first investigations which examined the influence of smoking on PAL in younger adolescents, and likely the first one in Southeastern Europe. Knowing the negative trends in PAL and alarmingly high prevalence of smoking in adolescents from the region, this is important strength of the study. Furthermore, usage of the previously validated questionnaires allowed us to objectively compare the obtained results with those previously reported for a similar sample. Therefore, we believe that, although not the final word, the investigation will contribute to knowledge and initiate further research.

## 5. Conclusions

Authors believe that detailed investigation of PA should be one of the priorities for optimizing health and productivity on the global level, given the fact that PA has a wide variety of positive outcomes, while inadequate PALs is one of the leading risk factors for global mortality. Unfortunately, regardless of the WHO appeal interventions aimed at increasing PALs that are still not prioritized on the global level, which is very clear given the fact that the decline in PALs is an ongoing trend. Hence, it is logical that a detailed investigation of PA and PA issues should also be an ongoing trend and highly prioritized. Investigations should proceed with exposing PA benefits/dangers of physical inactivity and identification of factors influencing PA levels since they present the most important preconditions for creating targeted and successful interventions aimed at fighting this negative trend, which took pandemic proportions. 

Future investigations should provide additional information that will finally encourage changes of global populations PALs and bring more concrete and quality solutions for solving the problem of physical inactivity. We believe that our results support this by providing information for future research and interventions since it provides insight and, thus, a better understanding of the relationships between smoking and PAL on the territory of Southeastern Europe. More specifically, the correlation between smoking and PAL in girls is promising from the viewpoint of development of public-health educational politics. Specifically, it may be suggested that the increase of the PAL in this age group may have multiple positive effects, including those directly related to higher PAL (i.e., prevention of non-communicable diseases, positive influence of PAL on psycho-social health), but also on prevention of smoking.

Smoking was not correlated to PAL in boys, but this finding is understandable considering the broader socio-cultural perspective. It seems that the socio-cultural environment in a studied country, as well as reported orientation of boys toward competitive sports, resulted in a non-systematic relationship between smoking and PAL. However, for more detailed analysis, further studies are needed, where different facets of PAL (i.e., physical education, free-time activities in nature, recreational activities in fitness centers, and competitive sports) will be specifically examined and correlated to PAL.

Finally, our results suggest that boys and girls in younger adolescence do not follow the equal trajectories when it comes to the relationships between smoking and PAL. Therefore, in developing promotional public health actions related to a decrease of smoking and increase of PAL, a gender-specific approach is highly recommended.

## Figures and Tables

**Figure 1 medicina-57-00214-f001:**
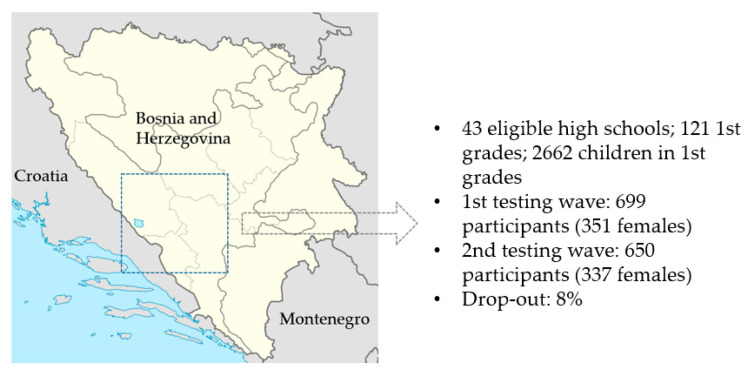
Study location, eligible participants, and number of participants tested at each testing wave.

**Figure 2 medicina-57-00214-f002:**
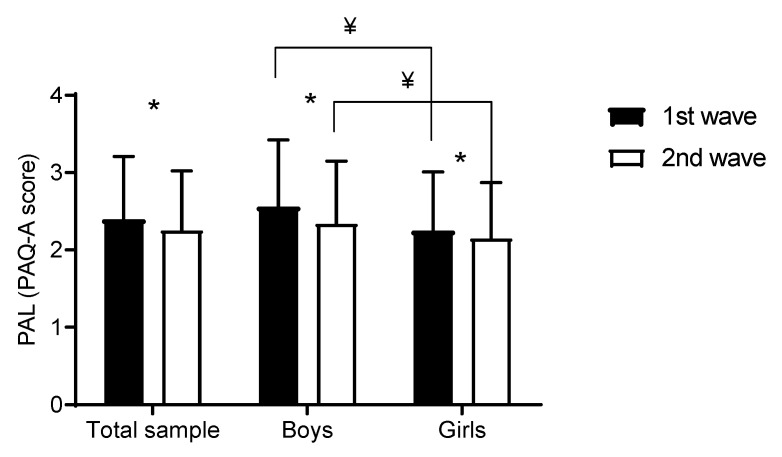
Changes and difference in physical activity levels for studied adolescents (¥ denotes significant (*p* < 0.05) *t*-test calculated between groups. * denotes significant (*p* < 0.05) *t*-test calculated within groups).

**Table 1 medicina-57-00214-t001:** Descriptive statistics (F—frequency, %—percentage), and differences (Chi square test—χ^2^) between adolescents according to their physical activity levels at the first-wave.

	Total		Insufficient PAL		Sufficient PAL		χ^2^	
	F	%	F	%	F	%	χ^2^	*p*
Gender							7.55	0.001
Boys	313	48.2	192	44.3	121	55.8		
Girls	337	51.8	241	55.7	96	44.2		
Socioeconomic status							1.01	4.47
Below average	17	2.6	9	2.1	8	3.8		
Average	560	86.3	372	85.3	188	88.3		
Above average	72	11.1	55	12.6	17	8		
Urban/Rural							0.17	0.67
Urban	296	45.6	200	46.2	96	44.2		
Rural	353	55.4	233	53.8	120	55.3		
Smoking							1.28	0.25
No	523	80.6	340	79	183	82.8		
Yes	127	19.4	89	20.6	38	17.2		

**Table 2 medicina-57-00214-t002:** Descriptive statistics (F—frequency, %—percentage), and differences (Chi square test—χ^2^) between adolescents according to their physical activity levels at the second-wave.

	Insufficient PAL		Sufficient PAL		χ^2^	
	F	%	F	%	χ^2^	*p*
Gender					7.54	0.001
Boys	192	44.3	121	55.8		
Girls	241	55.7	96	44.2		
Socioeconomic status					1.01	0.6
Below average	13	2.7	4	2.3		
Average	406	85.1	154	8.9		
Above average	56	11.7	16	9.1		
Urban/Rural					0.48	0.48
Urban	141	46.5	152	43.8		
Rural	162	53.5	195	56.2		
Smoking					6.46	0.02
No	231	76.2	292	84.2		
Yes	71	23.5	54	15.6		

**Table 3 medicina-57-00214-t003:** Logistic regressions between sociodemographic variables and cigarette smoking (predictors) and physical activity levels at the first-wave.

	Total Sample *		Gender Stratified			
			Boys		Girls	
	OR	95% CI	OR	95% CI	OR	95% CI
Cigarette smoking	0.55	0.36–0.83	0.76	0.50–1.01	0.75	0.55–0.95
Urban environment	0.97	0.75–1.23	1.09	0.81–1.42	0.95	0.67–1.36
Age	0.74	0.41–1.14	0.7	0.41–1.03	0.73	0.40–1.10
Socioeconomic status ^int^	0.99	0.81–1.14	1.00	0.71–1.29	0.99	0.75–1.24

* Model included “Gender” as the confounding factor, ^int^—observed as the interval for the purpose of logistic regression calculation (below average-average-above average socioeconomic status).

**Table 4 medicina-57-00214-t004:** Logistic regressions between sociodemographic variables and cigarette smoking (predictors) and physical activity levels at the second-wave.

	Total Sample *		Gender Stratified			
			Boys		Girls	
	OR	95% CI	OR	95% CI	OR	95% CI
Cigarette smoking	0.59	0.37–0.92	0.79	0.54–1.03	0.73	0.51–0.96
Urban environment	1.00	0.79–1.23	1.11	0.80–1.46	0.98	0.64–1.40
Age	0.80	0.40–1.16	0.65	0.38–1.07	0.76	0.36–1.08
Socioeconomic status ^int^	1.01	0.81–1.17	1.02	0.70–1.31	1.01	0.70–1.31

* Model included “Gender” as a confounding factor, ^int^—observed as an interval for the purpose of a logistic regression calculation (below average-average-above average socioeconomic status).

## Data Availability

The data presented in this study are available on request from the corresponding author. The data are not publicly available due to the founder’s policy.

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
