# Peer review of "Analysis of the Relationship between Tobacco Smoking and Physical Activity in Adolescence: A Gender Specific Study"

_medicina, 2021, doi:10.3390/medicina57030214_

Round 1

Reviewer 1 Report

Methods

Dear authors,

The manuscript Prospective analysis of the relationship between tobacco smoking and physical activity levels in early adolescence: a gender specific study is the important topic to examine.

I find it quite well written. Nerveless, please find a few comments below.

Methods

Please describe gender distribution among study participants in percent.

I think, the examples of items in the questionnaires could be provided.

line 144 "whan" change into "when"

Results

Have the authors considered to examine the predictors of change of smoking: predicting change from non-smoking to smoking from socio-demographic and physical activity variables?

Author Response

Please find responses in the attached file.

Thank you in advance

Authors

Reviewer 2 Report

This is a nice paper on the influence of the prospective relationship
between tobacco smoking and physical activity levels in early
adolescence. Strengths of the paper are the longitudinal design (with
follow-up after 20 months), the large sample size, and the investigation
of gender-specific effects. The paper is well written and the results of
interest. Please find below my remarks that may be helpful in further
improving the manuscript.

1. The paper is based on self-report data. Please discuss this important
limitation in more detail.

2. PAL seems to be analyzed as dichotomous outcome. Why not as
continuous variable?

3. In general, a detailed investigation of PA is recommendable. Please
discuss this issue in more detail.

4. Besides habitual smoking, a more detailed evaluation of smoking
history is needed. Please discuss.

5. The authors may want to discuss in more detail the applied
implications of the findings.

6. The results need to be put in context. What is the pattern in older
adolescents (based on prior evidence, authors expectations)? If there
are differences to present study, they need to be discussed in more detail.

7. What about other societies/geographic regions? Do authors expect
differences?

8. How can the findings observed support public health?

Author Response

Please find responses in the attached file.

Thank you.

Round 2

Reviewer 2 Report

We do not have further considerations about the paper. 

Author Response

We do not have further considerations about the paper. 

RESPONSE: Thank you for recognizing our efforts.